# EGOCENTRIC SPATIAL MEMORY NETWORK

## ABSTRACT

Inspired by neurophysiological discoveries of navigation cells in the mammalian brain, we introduce the first deep neural network architecture for modeling Egocentric Spatial Memory (ESM). It learns to estimate the pose of the agent and progressively construct top-down 2D global maps from egocentric views in a spatially extended environment. During the exploration, our proposed ESM network model updates belief of the global map based on local observations using a recurrent neural network. It also augments the local mapping with a novel external memory to encode and store latent representations of the visited places based on their corresponding locations in the egocentric coordinate. This enables the agents to perform loop closure and mapping correction. This work contributes in the following aspects: first, our proposed ESM network provides an accurate mapping ability which is vitally important for embodied agents to navigate to goal locations. In the experiments, we demonstrate the functionalities of the ESM network in random walks in complicated 3D mazes by comparing with several competitive baselines and state-of-the-art Simultaneous Localization and Mapping (SLAM) algorithms. Secondly, we faithfully hypothesize the functionality and the working mechanism of navigation cells in the brain. Comprehensive analysis of our model suggests the essential role of individual modules in our proposed architecture and demonstrates efficiency of communications among these modules. We hope this work would advance research in the collaboration and communications over both fields of computer science and computational neuroscience.

## 1 INTRODUCTION

Egocentric spatial memory (ESM) refers to a memory system that encodes, stores, recognizes and recalls the spatial information about the environment from an egocentric perspective Madl et al. (2015). Such information is vitally important for embodied agents to construct spatial maps and reach goal locations in navigation tasks.

For the past decades, a wealth of neurophysiological results have shed lights on the underlying neural mechanisms of ESM in mammalian brains. Mostly through single-cell electrophysiological recordings studies in mammals Madl et al. (2014), there are four types of cells identified as specialized for processing spatial information: head-direction cells (HDC), border and boundary vector cells (BVC), place cells (PC) and grid cells (GC). Their functionalities are: (1) According to Taube (2007), HDC, together with view cells Ekstrom et al. (2003), fires whenever the mammal's head orients in certain directions. (2) The firing behavior of BVC depends on the proximity to environmental boundaries Lever et al. (2009) and directions relative to the mammals' heads Burgess (2008). (3) PC resides in hippocampus and increases firing rates when the animal is in specific locations independent of head orientations Burgess (2008). (4) GC, as a metric of space Rowland et al. (2016), are regularly distributed in a grid across the environment Hafting et al. (2005). They are updated based on animal's speed and orientation Burgess (2008). The corporation of these cell types enables mammals to navigate and reach goal locations in complex environments; hence, we are motivated to endow artificial agents with the similar memory capability but a computational architecture for such ESM is still absent.

Inspired by neurophysiological discoveries, we propose the first computational architecture, named as the Egocentric Spatial Memory Network (ESMN), for modeling ESM using a deep neural network. ESMN unifies functionalities of different navigation cells within one end-to-end trainable framework

and accurately constructs top-down 2D global maps from egocentric views. To our best knowledge, we are the first to encapsulate the four cell types respectively with functionally similar neural network-based modules within one integrated architecture. In navigation tasks, the agent with the ESMN takes one egomotion from a discrete set of macro-actions. ESMN fuses the observations from the agent over time and produces a top-down 2D local map using a recurrent neural network. In order to align the spatial information at the current step with all the past predicted local maps, ESMN estimates the agent's egomotion and transforms all the past information using a spatial transformer neural network. ESMN also augments the local mapping module with a novel spatial memory capable of integrating local maps into global maps and storing the discriminative representations of the visited places. The loop closure component will then detect whether the current place was visited by comparing its observation with the representations in the external memory which subsequently contributes to global map correction.

Neuroscience-inspired AI is an emerging research field Hassabis et al. (2017). Our novel deep learning architecture to model ESMN in the mammalian navigation system attempts to narrow the gap between computer science (CS) and computational neuroscience (CN) and bring interests to both communities. On one hand, our novel ESMN outperforms several competitive baselines and the state-of-the-art monocular visual SLAMs. Our outstanding performance in map construction brings great advancements in robotics and CS. It could also have many potential engineering applications, such as path planning for robots. (2) In CN, the neuroplausible navigation system with four types of cells integrated is still under development. In our work, we put forward bold hypothesis about how these navigation cells may cooperate and perform integrated navigation functions. We also faithfully propose several possible communication links among them in the form of deep architectures.

We evaluate ESMN in eight 3D maze environments where they feature complex geometry, varieties of textures, and variant lighting conditions. In the experiments, we demonstrate the acquired skills of ESMN in terms of positional inference, free space prediction, loop closure classification and map correction which play important roles in navigation. We provide detailed analysis of each module in ESMN as well as their functional mappings with the four cell types. Lastly, we conduct ablation studies, compare with state-of-the-art Simultaneous Localization And Mapping (SLAM) algorithms and show the efficacy of our integrated framework on unifying the four modules.

## 2   RELATED WORKS

There is a rich literature on computational models of egocentric spatial memory (ESM) primarily in cognitive science and AI. For brevity, we focus on the related works in machine learning and robotics.

Reward-based learning is frequently observed in spatial memory experiments Hok et al. (2005). In machine learning, reinforcement learning is commonly used in attempts to mathematically formalize the reward-based learning. Deep Q-networks, one of the reinforcement learning frameworks, have been employed in the navigation tasks where the agent aims to maximize the rewards while navigating to the goal locations Mirowski et al. (2016); Mnih et al. (2015; 2016). The representation of spatial memory is expressed implicitly using long short term memory (LSTM) Hochreiter & Schmidhuber (1997). In one of the most relevant works Gupta et al. (2017), Gupta *et al.* introduced a mapper-planner pipeline where the agent is trained in a supervised way to produce a top-down belief map of the world and thus plans path; however, the authors assume that the agents perform mapping tasks in an ideal scenario where the agent knows which macro-action to take and takes the macro-action without control errors at each time step. Different from their work, our ESM network takes into account the action uncertainty and predicts the agent's pose based on egocentric views. Moreover, we propose the mechanism of loop closure and map correction for long-term exploration.

Apart from reinforcement learning, there are works on navigation in the domain of robotics where the spatial memory is often explicitly represented in the form of grid maps or topological structuresElfes (1987); Peasley et al. (2012); Kuipers & Byun (1991). SLAM tackles the problem of positional inference and mapping Smith et al. (2013); Durrant-Whyte et al. (2003); Dissanayake et al. (2000); Thrun et al. (2002). While classical SLAM achieves good performances with the aid of multimodal sensors Lenac et al. (2017), SLAM using monocular cameras still has limitations where the feature matching process relies heavily on hand-crafted features extracted from the visual scenes Zhang & Vela (2015); Pollefeys et al. (1999). As great strides have been made using deep learning in computer vision tasks which results in a significant performance boost, several works endeavor to replace parts

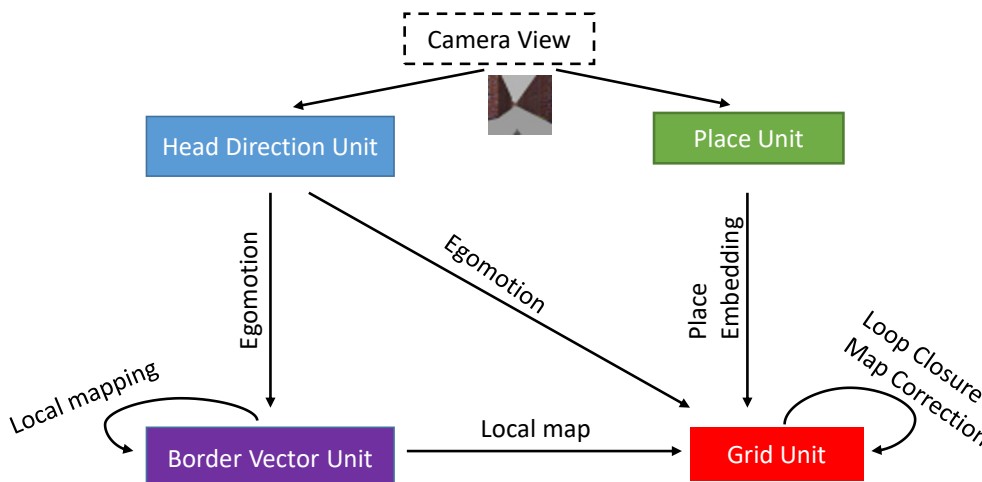

Figure 1: Overview of our proposed Egocentric Spatial Memory Network. It consists of Head Direction Unit, Boundary Vector Unit, Place Unit, and Grid Unit. See Section 3.1 for more details.

of the classical SLAM workflows with deep learning based modules Kendall et al. (2015); Agrawal et al. (2015); Kendall & Cipolla (2016). However, there is no existing end-to-end neural network for visual SLAM to our best knowledge.

Inspired by the neurophysiological results, we propose the first unified ESM model using deep neural networks. This model simultaneously solves the problems of positional inference, intrinsic representation of places and space geometry during the process of 2D map construction.

## 3 PROPOSED ARCHITECTURE

Before we elaborate on our proposed architecture, we introduce the Egocentric Spatial Memory Network (ESMN) modeling problem and relevant notations. ESM involves an object-to-self representational system which constantly requires encoding, transforming and integrating spatial information from the first-person view into a global map. At the current time step $t$, the agent, equipped with one RGB camera, takes the camera view $I_t$ as the current visual input. Without the aid of any other sensors, it predicts the egomotion from the pair of camera views $(I_{t-1}, I_t)$. At time step $t$, the agent is allowed to take only one egomotion $A_{\theta,d,l,t}$ out of a discrete set of macro-actions which include rotating left/right by $\theta$ degrees and moving in the directions $l$: forward/backward/left/right by the distance of $d$ relative to the agent's current pose. We assume the agent moves in a 2D grid world and the camera coordinate is fixed with respect to the agent's body. The starting location $p_0$ of the agent is always $(0, 0, 0)$ where the triplet denotes the positions along the x and y-axis and the orientation in the world coordinate. The problem of modeling ESM is to learn a global map in the egocentric coordinate based on the visual input pairs $(I_{t-1}, I_t)$ for $t = 1, \ldots, T$. We define the global map as a top-view 2D probabilistic grid map where the probability infers the agent's belief of free space. For precise controls of the experimental variables, we consider the egocentric spatial memory (ESM) modeling problem in artificial 3D maze environments. In order to tackle this problem, we propose a unified neural network architecture named as ESMN.

### 3.1 OVERVIEW

The architecture of ESMN is illustrated in Figure 1 and is elaborated in details in Figure 2. Inspired by the navigation cells mentioned in the introduction, our proposed ESMN comprises four modules: Head Direction Unit (HDU), Boundary Vector Unit (BVU), Place Unit (PU), and Grid Unit (GU). This is to incorporate multiple objectives for modeling ESM. (1) HDU learns to estimate the egomotion at each time step by minimizing the classification errors with a 2D convolution neural network (2D-CNN). (2) BVU serves as a local mapper and predicts 2D top-view local maps representing free space. It minimizes errors of predicted free space in the local map by using a recurrent neural network. Based on the estimated egomotion, a spatial transformer module transforms all the past

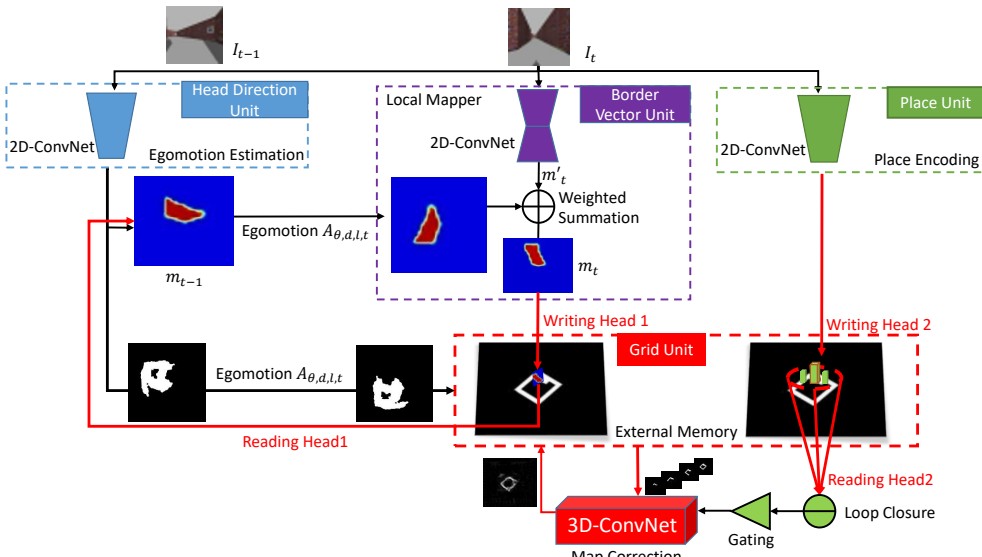

Figure 2: Architecture of our proposed Egocentric Spatial Memory Network. See Section 3.1 for the overview of individual module.

spatial information to the current egocentric coordinate via a 2D-CNN. (3) PU learns to encode the latent representation of visited places in a 2D-CNN pre-trained using a triplet loss. (4) GU, as a memory, integrates the predicted local maps from BVU over time and keeps track of all the visited places. In GU, we add a gating mechanism to constantly detect the loop closure and thus eliminate the accumulated errors during long-term mapping.

In this paper, we adopt a "divide and conquer" approach by composing different navigation modules within one framework systematically. The division of these modules are inspired by scientific supports which have shown the advantages in biological systems Murray et al. (2016). Leveraging on rich features extracted from deep networks, the learnt features are suitable for recognizing visual scenes and hence, boost up the map construction and correction performances. The efficiency and robustness of our algorithm are demonstrated by comparing with other spatial navigation methods replied on visual inputs.

## 3.2 HEAD DIRECTION UNIT (HDU): POSITIONAL INFERENCE

At each time step, the RGB camera image (or frame) is the only input to the agent. In order to make spatial reasoning about the topological structure of the spatially extended environment, the agent has to learn to take actions to explore its surroundings and predict their own poses by integrating the egomotion over time. The HDU module in our proposed ESMN model employs a 2D-CNN to predict the macro-action the agent has taken at the time step $t$ based on the inputs of two camera views $\{I_{t-1}, I_t\}$. Though there are two possible implementations for the loss in training the HDU module, i.e. regression and classification, we consider the latter in order to reduce the dimensionality of action space. The macro-actions $A_{\theta,d,l}$ are discretized into 2 classes for rotation and 4 classes for translation as explained at the beginning of Section 3. Though we solve the problem of positional inference using a feed-forward neural network based on two most recent observations alone, the extensions to use recurrent neural network would be interesting to explore in the future.

## 3.3 BOUNDARY VECTOR UNIT (BVU): LOCAL MAPPER

We explain how ESMN integrates egocentric views into a top-down 2D representation of the environment using a recurrent neural network. Similar to Gupta et al. (2017), the BVU in ESMN serves as a local mapper and maintains the accumulative free space representations in the egocentric coordinate for a short-term period. Given the current observation $I_t$, function $g$ first encodes its geometric representation about space $g(I_t)$ via a 2D-CNN and then transform $g(I_t)$ into egocentric top-down view $m'_t$ via de-convolutional layers. Together with the accumulative space representation $m_{t-1}$ at the previous step and the estimated egomotion $A_{\theta,d,l,t}$ from $t-1$ to $t$, BVU estimates the current

local space representation $m_t$ using the following update rule:

$$m_t = U(W(m_{t-1}, A_{\theta,d,l,t}), m_t'),$$ (1)

where $W$ is a function that transforms the previous accumulative space representation $m_{t-1}$ to the current egocentric coordinate based on the estimated egomotion $A_{\theta,d,l,t}$. We parameterize $W$ by using a spatial transformer network Jaderberg et al. (2015) composing of two key elements: (1) it generates the sampling grid which maps the input coordinates of $m_{t-1}$ to the corresponding output locations after egomotion $A_{\theta,d,l,t}$ transformation; (2) the sampling kernel then takes the bilinear interpolation of the values on $m_{t-1}$ and outputs the transformed $m_{t-1}$ in the current egocentric coordinate. $U$ is a function which merges the free space prediction $m_t'$ from the current observation with the accumulative free space representation at the previous step. Specifically, we simplify merging function $U$ as a weighted summation parameterized by $\lambda$ followed by hyperbolic tangent operation:

$$U(W(m_{t-1}, A_{\theta,d,l,t}), m_t') = \frac{e^{2(\lambda m_t' + (1-\lambda)W(m_{t-1}, A_{\theta,d,l,t}))} - 1}{e^{2(\lambda m_t' + (1-\lambda)W(m_{t-1}, A_{\theta,d,l,t}))} + 1}.$$ (2)

### 3.4 PLACE UNIT (PU): PLACE ENCODING AND LOOP CLOSURE CLASSIFICATION

Loop closure is valuable for the agents in the navigation tasks as it encourages efficient exploration and spatial reasoning. In order to detect loop closure during an episode, given the current observation $I_t$, ESMN learns to encode the discriminative representation $h(I_t)$ of specific places independent of scaling and orientations via an embedding function $F$. Based on the similarity of all the past observations $\Omega_t = \{I_1, I_2, ..., I_t\}$ at corresponding locations $P_t = \{p_1, p_2, ..., p_t\}$, we create training targets by making an analogy to the image retrieval tasks Gordo et al. (2016) and define the triplet $(I_t, I_+, I_-)$ as anchor sample (current camera view), positive and negative samples drawn from $\Omega_t$ respectively. PU tries to minimize the triplet loss:

$$L_{triplet}(F(I_t, I_+, I_-)) = -\log \frac{e^{-D(F(I_t, I_+))}}{e^{-D(F(I_t, I_+))} + e^{-D(F(I_t, I_-))}},$$ (3)

where we parameterize $F$ using a three-stream 2D-CNN where the weights are shared across streams. $D$ is a distance measure between pairs of embedding. Here, mean squared error is used. A loop closure label equals 1 if the mean squared error is below the threshold $\alpha$ which is set empirically.

Apart from the similarity comparison of observations, we consider two extra criteria for determining whether the place was visited: (1) the current position of the agent $p_t$ is near to the positions visited at the earlier times. We empirically set a threshold distance between the current and the recently visited locations based on the loop closure accuracy during training. We implemented it via a binary mask in the center of the egocentric global map where the binary states denote the accepted "closeness". (2) the agent only compares those positions which are far from the most recent visited positions to avoid trivial loop closure detection at consecutive time steps. It is implemented via another binary mask which tracks these recently visited places. These criterion largely reduce the false alarm rate and improves the searching speed during loop closure classification.

### 3.5 GRID UNIT (GU): GLOBAL MAPPER AND PLACE TRACKING

While BVU provides accumulative local free space representations in high resolution for a short-term period, we augment the local mapping framework with memory for long-term integration of the local maps and storage of location representations. Different from Neural Turing Machine Graves et al. (2014) where the memory slots are arranged sequentially, our addressable memory, of size $F \times H \times W$, is indexed by spatial coordinates $\{(i,j) : i \in \{1, 2, ..., H\}, j \in \{1, 2, ..., W\}\}$ with memory vector $M(i,j)$ of size $F$ at location $(i,j)$. Because ESM is often expressed in the coordinate frame of the agent itself, we use location-based addressing mechanism and the locations of reading or writing heads are fixed to be always in the center of the memory. Same as BVU, all the past spatial information in the memory is transformed based on the estimated egomotion $A_{\theta,d,l,t}$. Mathematically, we formulate the returned reading vector $r_{h,w}$ as

$$r_{h,w} = \left\{ M(i,j) : i \in \left\{\frac{H}{2} - \frac{h}{2}, ..., \frac{H}{2} + \frac{h}{2}\right\}, j \in \left\{\frac{W}{2} - \frac{w}{2}, ..., \frac{W}{2} + \frac{w}{2}\right\} \right\}$$ (4)

where the memory patch covers the area of memory vectors with the width $w$ and height $h$. We simplify the writing mechanism for GU and use Equation 4 for writing the vector $r_{h,w}$.

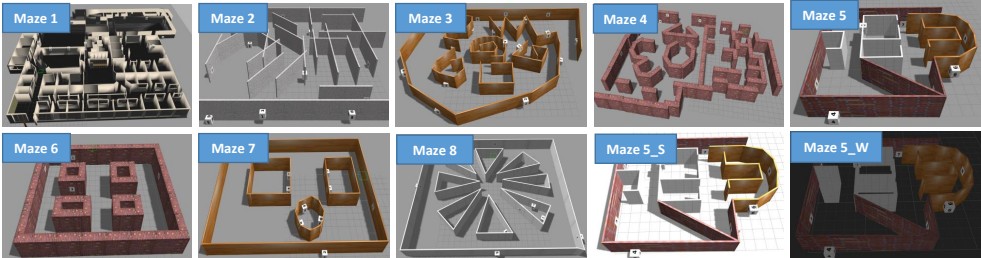

Figure 3: Overview of maze layouts, with differing geometries, textures and lighting conditions. Each maze is stimulated with normal, weak and strong lighting conditions. Maze 5_S and 5_W refers to Maze 5 with strong and weak lighting conditions respectively. Maze 1 is adopted from Willow Garage Garage (2011). Maze 8 is inspired by radial arm maze used to test spatial memory in rats Olton & Samuelson (1976). The digits are pasted on walls along specific pathways for loop closure classification tasks.

In our case, two writing heads and two reading heads are necessary to achieve the following objectives: (1) one reading head returns the memory vectors $m_{t-1}$ in order for BVU to predict $m_t$ using Equation 1; (2) GU performs updates by writing the predicted local accumulative space representation $m_t$ back into the memory to construct the global map in the egocentric coordinate; (3) GU keeps track of the visited places by writing the discriminative representation $h(I_t)$ at the center of the egocentric global map denoted as $(\frac{H}{2}, \frac{W}{2})$; (4) GU returns the memory vectors near to the current location for loop closure classification where the size of the searching area is parameterized by $w$ and $h$. The choice of $w$ and $h$ are application-based. We simplify the interaction between local space representations $m_t$ and $m_{t-1}$ with GU and set $w$ and $h$ to be the same size as $m_t$ and $m_{t-1}$.

If the loop closure classifies the current observation as "visited", GU eliminates the discrepancies on the global map by merging the two places together. The corrected map has to preserve the topological structure in the discovered areas and ensure the connectivity of the different parts on the global map is maintained. To realize this, we take three inputs in a stack of 3D convolution and de-convolution layers for map correction. The inputs are: (1) the local map predicted at the anchor place; (2) the local map predicted at the recalled place in the current egocentric coordinate; (3) all the past integrated global maps. To make the training targets, we perturb the sequence of ground truth egomotions with random ones and generate synthetic integrated global maps with rotation and scaling augmentations. We minimize regression loss between the predicted maps and the ground truth.

### 3.6 TRAINING AND IMPLEMENTATION DETAILS

We train ESMN end-to-end by stochastic gradient descent with learning rate 0.002 and momentum 0.5. Adam Optimizer Kingma & Ba (2014) is used. At each time step, the ground truths are provided: local map, egomotion and loop closure classification label. For faster training convergence, we first train each module separately and then load these pre-trained networks into ESMN for fine-tuning. The input frame size is $3 \times 64 \times 64$. We normalize all the input RGB images to be within the range $[-1, 1]$. The batch size is 10. The discriminative representation $h(I_t)$ is of dimension 128. The size of the local space representation $m_t$ is $h \times w = 32 \times 32$ covering the area $7.68\ meters \times 7.68\ meters$ in the physical world whereas the global map (the addressable memory) is of size $H \times W = 500 \times 500$. We set $\lambda = 0.5$ in BVU and $\alpha = 0.01$ in PU. The memory vector in GU is of size $F = 128$. Refer to Supplementary Material for detailed architecture.

## 4 EXPERIMENTS

We conduct experiments on eight 3D mazes in Gazebo for robotic simulation Koenig & Howard (2004). We will make our simulation source code public once our paper is accepted. Figure 3 shows an overview of eight mazes. They feature complex geometry and rich varieties of textures. To prevent over-fitting in a single-modality environment, we augment these observations with different illumination conditions classified into three categories: strong, normal and weak, and train our ESMN in these mixed lighting conditions. Based on the results in Section 4.1, our ESMN performs equally well in the test set across permutations of these conditions which shows its capacity of generalizing

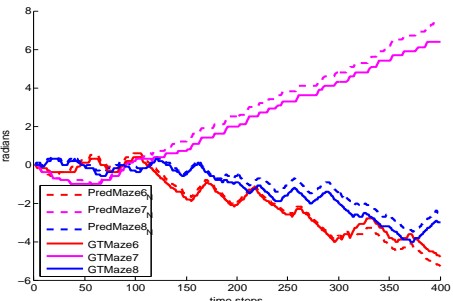

| | Weak | Normal | Strong |
|---|---|---|---|
| Maze 6 | | | |
| Rotation | 0.89 | 0.95 | 0.96 |
| Rotation + Translation | 0.87 | 0.94 | 0.93 |
| Maze 7 | | | |
| Rotation | 0.99 | 1 | 0.98 |
| Rotation + Translation | 0.90 | 0.89 | 0.89 |
| Maze 8 | | | |
| Rotation | 0.99 | 0.85 | 1 |
| Rotation + Translation | 0.95 | 0.84 | 0.95 |

Figure 4: Head direction prediction of three example paths in Maze 6,7,8 (normal lighting conditions) by accumulating the estimated egomotions across first 400 time steps. The dotted line is the prediction and the solid is the ground truth.

Table 1: Egomotion Classification Accuracy in Maze 6,7,8 under *weak*, *normal*, and *strong* lighting conditions across first 400 time steps.

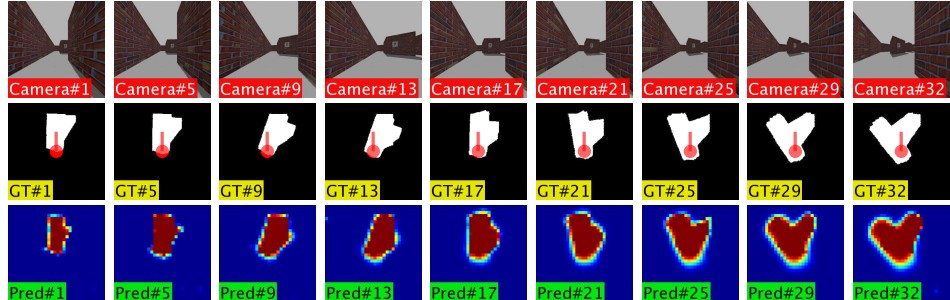

Figure 5: Example results of predicted local maps over first 32 time steps in Maze 6. Every 4 out of 32 frames are shown (left to right columns). Row 1 shows the camera views. Row 2 shows the ground truth with red arrows denoting the agent's position and orientation from the top view. The white region denotes free space while the black denotes unknown areas. Row 3 shows the corresponding top-view predicted local maps where the red color denotes higher belief of the free space.

in various environments. For loop closure classification tasks, we create digits on walls as the unique features of specific pathways. The agent randomly walks through the mazes with obstacle avoidance. The ground truths for 2D top-view of mazes are obtained by using one virtual 2D laser scanner attached to the agent. We use the data collected in maze 1 to 5 for training and validation and the data in maze 6 to 8 for testing. The macro-actions for egomotions are divided into 2 classes for rotation by $\theta = 10\ degrees$ and 4 classes for translation with $d = 0.1\ meters$ relative to the agent's current pose in the physical world. The simulation environment enables us to conduct exhaustive evaluation experiments and train individual module separately using fully supervised signals for better functionality evaluation. The integrated framework is then trained jointly after loading the pre-trained weights from individual modules.

## 4.1 EVALUATIONS OF INDIVIDUAL MODULES

**Head Direction Unit (HDU): Positional Inference.** Head Direction Cells fire whenever the animal's head orients in certain directions. The study McNaughton et al. (2006) suggests that the rodent's orientation system is as a result of neural integration of head angular velocities derived from vestibular system. Thus, after jointly training ESMN, we decode HDU for predicting head direction in the world coordinate by accumulating the estimated egomotions over time. Figure 4 shows the head direction prediction in the world coordinate compared with the agent's ground truth global poses. We observe that the head directions integrated from the estimated egomotions are very similar to the ground truths. Moreover, we compute the egomotion classification errors from 2 classes of rotation and 4 classes of translation under three illumination conditions, see Table 1. There is a slight degradation of rotation + translation compared with rotation alone. One possible reason is that the change of egocentric camera views after 10 degrees of rotation is more observable than the one induced from translation by 0.1 meters.

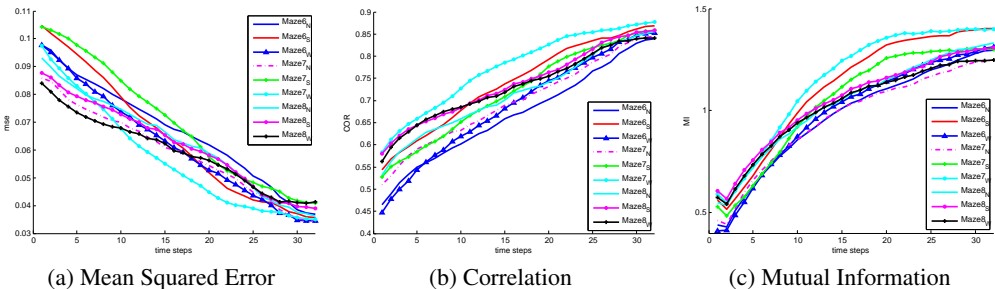

|                      |                  |                        |
| :------------------: | :--------------: | :--------------------: |
| (a) Mean Squared Error | (b) Correlation | (c) Mutual Information |

Figure 6: Evaluation of Local Mapper using Mean Squared Error (MSE), Correlation, Mutual Information (MI) across first 32 time steps in Maze 6, 7, 8 under normal (N), weak (W), and strong (S) illumination conditions. The predicted local maps are compared with the ground truth at $t = 32$. Smaller is better for MSE. Larger is better for correlation and MI. Best viewed in color.

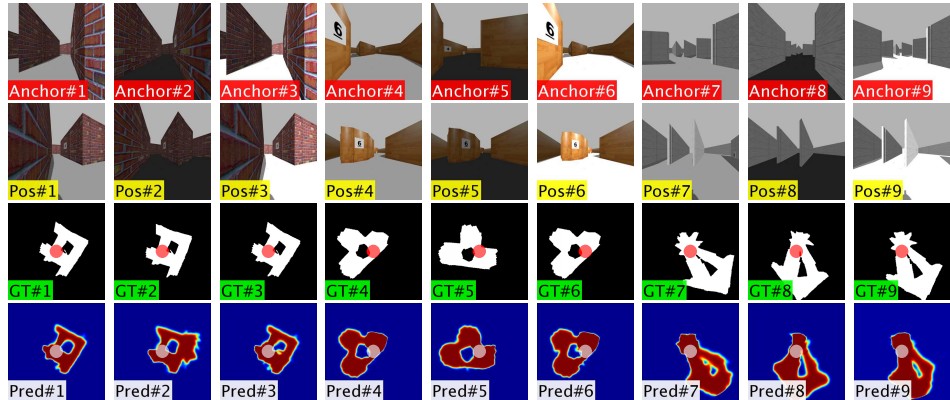

Figure 7: Example observation pairs when the loop closure is detected. Row 1 are the anchors (current camera views). Row 2 are the camera views from the previously visited places where the loop closure is detected. Row 3 show the agent's locations (red circle) on the ground truth maps. Row 4 show the agent's locations (white circle) on the predicted map with ground truth poses.

**Boundary Vector Unit (BVU): Local Mapper.** The firing behavior of Border Cell and Boundary Vector Cells depends on the proximity to environmental boundaries Lever et al. (2009) as well as the directions relative to the mammals' heads Burgess (2008). In Section 3.3, we express the proximity to physical obstacles by predicting 2D top-view free space representations and accumulate the local belief maps over time based on the egomotions. Figure 5 shows example results of predicted accumulative free space representations over 32 time steps in Maze 6. See the Supplementary Material for more examples.

We also provide quantitative analysis of our predicted local maps across the first 32 time steps in Figure 6 using Mean Squared Error (MSE), Correlation (Cor), Mutual Information (MI) which are standard image similarity metrics Mitchell (2010). At each time step, the predicted local maps are compared with the ground truth maps at $t = 32$. As the agent continues to explore in the environment, the area of the predicted free space expands leading to the decrease of MSE and the increase of correlation and MI in our test set. This validates that ESMN is able to accurately estimate the proximity to the physical obstacles relative to the agent itself and continuously accumulate the belief of free space representations based on the egomotions.

**Place Unit (PU): Loop Closure Classification.** One of the critical aspects of ESM is the ability to recall and recognize a previously visited place independent of head orientation. Place cell (PC) resides in hippocampus and increases firing patterns when the animal is in specific locationsBurgess (2008). Recent anatomical connectivity research Solstad et al. (2006) implies that grid cells (GC) are the principle cortical inputs to PC. In our proposed ESMN, we design a novel memory unit (GU) for storing the discriminative representations of visited places based on their relative locations with respect to the agent itself in the egocentric coordinate. GU interacts constantly with PU by reading and writing the memory vectors. We evaluate the learnt discriminative representations of visited

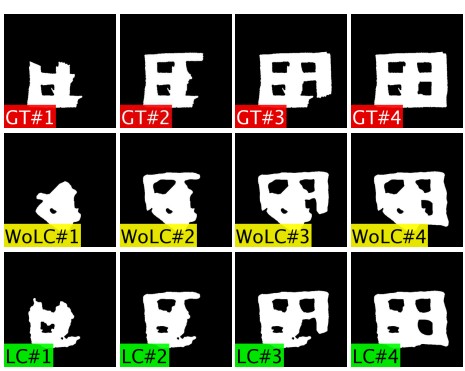

|  |  | MSE | Cor | MI |
|---|---|---|---|---|
| Map Evaluation at $t = 448$ | | | | |
| base-lines | 3D-CNN | 0.09 | 0.50 | 0.19 |
| | LSTM Direct | 0.09 | 0.48 | 0.15 |
| ablated models | HDU + BVU | 0.06 | 0.67 | 0.28 |
| | HDU + BVU + PU + GU | **0.04** | **0.81** | **0.36** |
| Map Evaluation at $t = 1580$ | | | | |
| base-line | 3D-CNN | $t$ > model's capacity (480) | | |
| | LSTM Direct | 0.24 | 0.49 | 0.23 |
| ablated models | HDU + BVU | 0.06 | 0.83 | 0.58 |
| | HDU + BVU + PU + GU | **0.04** | **0.91** | **0.72** |

Figure 8: Example results of constructed global maps in the world coordinate in Maze 6 across 1580 time steps. The topmost row shows the ground truth. Row 2 and Row 3 show the corresponding top-view accumulative belief of the predicted global maps without and with loop closure classification at $t = 448$ respectively.

Table 2: Ablation study on the global map performance in Maze 6 at $t = 448$ and $t = 1580$ using metrics in Section 4.1. From top to bottom, the models are: 3D-CNN baseline, LSTM baseline, our ablated model with PU and GU removed, our proposed architecture. The best values are highlighted in bold.

places by comparing the current observation (anchor input) with all the past visited places. Figure 7 presents example pairs of observations when the loop closure is detected. Qualitative results imply that our ESMN can accurately detect loop closure when the agent is at the previously visited places irrespective of large differences between these camera views.

**Grid Unit: Mapping Integration and Correction.** After the loop closure is detected, ESMN performs map correction to eliminate the discrepancies during long-term mapping. Figure 8 presents the example results of the predicted global maps after map correction. It is shown that the map gets corrected at $t = 448$ (Column 1). Thus, the predicted global map (Row 3) is structurally more accurate compared with the one without loop closure (Row 2).

## 4.2 ABLATION ANALYSIS

Our ablation analysis is as follows: (1) To study the necessity of egomotion estimation from HDU, we take a sequence of camera views at all the previous time steps as inputs to predict the global map directly. We implement this by using a feed-forward 3D convolution neural network (3D-CNN). Practically, since it is hard to take all the past camera views across very long time period, we choose the input sequence with one representative frame every 15 time steps. (2) As ESM requires sustainable mapping over long durations, we create one more baseline by taking the sequence of camera views as inputs and using Long Short Term Memory architecture to predict global maps directly (LSTM Direct). To maintain the same model complexity, we attach the same 2D-CNN in our BVU module before LSTM and fully connected layers after LSTM. (3) To explore the effect of loop closure and thus map correction, we create one ablated model with PU and GU removed (HDU + BVU). (4) We present the results of our integrated architecture with loop closure classification and map correction enabled (HDU + BVU + PU + GU). We report the evaluation results in Table 2 using the metrics MSE, correlation and MI as introduced in Section 4.1.

We observe that our proposed architecture surpasses the competitive baselines and the ablated models. At $t = 448$, compared with the first baseline (3D-CNN), there is decrease of 0.03 in MSE and increase of 0.17 in correlation and 0.09 in MI. The significant improvement infers that it is necessary to estimate egomotion for better map construction. Additionally, the integration of local maps based on the egomotion makes the computation more flexible and efficient by feeding back the accumulative maps to the system for future time steps. In the second baseline (LSTM Direct), we observe that the performance drops significantly when it constructs global maps for longer durations. As GU serves as an external memory to integrate local maps, the baseline confirms GU has advantages over LSTMDirect in terms of long-lasting memory. To explore the effect of loop closure and thus map correction, we have the ablated model with PU and GU removed (HDU + BVU). Compared with

|  | Maze 6 | | | Maze 7 | | | Maze 8 | | |
|---|---|---|---|---|---|---|---|---|---|
| Metrics | MSE | Cor | MI | MSE | Cor | MI | MSE | Cor | MI |
| Ours | **0.04** | **0.81** | **0.36** | **0.06** | **0.73** | **0.32** | **0.14** | **0.53** | **0.21** |
| EFKslam | 0.10 | 0.65 | 0.25 | 0.10 | 0.70 | 0.31 | - | - | - |
| ORBslam | 0.12 | 0.41 | 0.09 | 0.15 | 0.28 | 0.05 | 0.24 | 0.17 | 0.02 |

Table 3: Comparison of our method with state-of-the-art SLAM algorithms. The global map results after the agent completes one loop closure in Maze 6, 7, and 8 are reported using metrics introduced in Section 4.1. The numbers highlighted in bold are the best.

our proposed architecture with all four modules enabled at $t = 448$ and $t = 1580$, the decreased performance validates that these steps are necessary to eliminate the errors during long-term mapping.

### 4.3 Comparison with monocular visual SLAMs

We evaluate our ESMN by comparing with state-of-the-art monocular visual SLAM methods, EK-Fslam Civera et al. (2010), and ORBslam Mur-Artal et al. (2015). We used the codes provided by the authors and replaced the intrinsic camera parameters with ours. In order to construct maps in the world scale, we provide explicit scaling factor for both methods. Results show that our ESMN significantly outperforms the rest in terms of MSE, Cor and MI as shown in Table 3. In the experiments, we also observe that ORBslam has high false positives in loop closure detection which leads to incorrect map correction. It may be caused by the limitations of local feature descriptors. Different from them, our ESMN can robustly detect loop closure and correct maps over time. Furthermore, we notice that both SLAMs tend to fail to track feature descriptors between frames especially during rotations. The parameters of searching windows in order to match features have to be manually set and they cannot be adapted in large motion cases. The results in Maze 8 using EKFslam are not provided in Table 3 as the original algorithm requires the dense local feature matching and it is not applicable in very low texture environments.

## 5 Conclusion

We get inspirations from neurophysiological discoveries and propose the first deep neural network architecture for modeling ESM which unifies the functionalities of the four navigation cell types: head-direction cells, border cells and boundary vector cells, place cells and grid cells. Our learnt model demonstrates the capacity of estimating the pose of the agent and constructing a top-down 2D spatial representations of the physical environments in the egocentric coordinate which could have many potential applications, such as path planning for robot agents. Our ESMN accumulates the belief about the free space by integrating egocentric views. To eliminate errors during mapping, ESMN also augments the local mapping module with an external spatial memory to keep track of the discriminative representations of the visited places for loop closure detection. We conduct exhaustive evaluation experiments by comparing our model with some competitive baselines and state-of-the-art SLAM algorithms. The experimental results demonstrate that our model surpasses all these methods. The comprehensive ablation study suggests the essential role of individual modules in our proposed architecture and the efficiency of communications among these modules.

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

# 6 SUPPLEMENTARY

## 6.1 EXAMPLE RESULTS OF BOUNDARY VECTOR UNIT (BVU) AS LOCAL MAPPER

We express the proximity to physical obstacles by predicting 2D top-view free space representations and accumulate the local belief maps over time based on the egomotions. Figure 9,10,11 show example results of predicted accumulative free space representations over 32 time steps in Maze 6, Maze 7 and Maze 8.

## 6.2 DETAILED ARCHITECTURES OF OUR PROPOSED EGOCENTRIC SPATIAL MEMORY NETWORK (ESMN)

We introduce the anatomy of our model for reproducibility in this section. We follow exactly the same convention as Torch. Inspired by neurophysilogical discoveries, our framework consists of four units: Head Direction Unit (HDU), Boundary Vector Unit (BVU), Place Unit (PU), and Grid Unit (GU). Refer to Table 7 for HDU, Table 6 for BVU, Table 8 for PU, Table 9 for global map correction.

In GU, as explained in Section 3.5 in the main manuscript, egocentric spatial memory is often expressed in the coordinate frame of the agent itself, we use location-based addressing mechanism and the locations of reading or writing heads are fixed to be always in the center of the external memory. As the receptive field parameterized by $w$ and $h$ for reading and writing are fixed. We designed a 2D binary masking map where 1 denotes the place

| Writing Head 1 | |
| --- | --- |
| binary masking map | local map at $t$ |
| nn.Identity() | nn.SpatialUpSamplingNearest(8) |
| | nn.Padding(3, 122, 4, 0) |
| | nn.Padding(3, -122, 4, 0) |
| | nn.Padding(4, -122, 4, 0) |
| | nn.Padding(4, 122, 4, 0) |
| nn.CMulTable() | |

| Reading Head 1 |
| --- |
| Global map at time $t$ |
| nn.Sequential() |
| nn.SplitTable(3,4) |
| nn.NarrowTable(123,256) |
| nn.JoinTable(2,3) |
| nn.Reshape(batchSize, 1, 256, 500) |
| nn.SplitTable(4,4) |
| nn.NarrowTable(123,256) |
| nn.JoinTable(2,3) |
| nn.Reshape(batchSize, 1, 256, 256) |
| downsample tensor by scale of 8 using Spatial Transformer Network |

Table 4: Reading Head 1 and Writing Head 1 in Grid Unit (GU). It either reads or writes the local map of size $32 \times 32$ from the global map of size $500 \times 500$ in GU.

| Writing Head 2 | |
| --- | --- |
| binary masking map | $h(I_t)$ at time step $t$ |
| nn.Identity() | nn.Replicate(8,3,4) |
| | nn.Reshape(batchSize,128,8,1) |
| | nn.Replicate(8,4,4) |
| | nn.Reshape(batchSize,128,8,8) |
| | nn.Padding(3, 246, 4, 0) |
| | nn.Padding(3, -246, 4, 0) |
| | nn.Padding(4, -246, 4, 0) |
| | nn.Padding(4, 246, 4, 0) |
| nn.CMulTable() | |

| Reading Head 2 |
| --- |
| Global map storing all spatial representations at time $t$ |
| nn.Sequential() |
| nn.Reshape(batchSize, 1, 128, 500, 500) |
| nn.SplitTable(4,5) |
| nn.NarrowTable(123,256) |
| nn.JoinTable(2,4) |
| nn.Reshape(batchSize, 1, 128, 256, 500) |
| nn.SplitTable(5,5) |
| nn.NarrowTable(123,256) |
| nn.JoinTable(2,4) |
| nn.Reshape(batchSize, 128, 256, 256) |
| downsample tensor by scale of 8 using Spatial Transformer Network |

Table 5: Reading Head 2 and Writing Head 2 in Grid Unit (GU). It either reads memory vectors or writes the discriminative spatial representation $h(I_t)$ of size $1 \times 128$ from or into the global map of size $128 \times 500 \times 500$ in GU.

to read and write and vice versa. We use element-wise multiplication between each feature map of the external memory and the binary masking map for reading and writing operations. Refer to Table 4 for reading and writing local maps in GU, and Table 5 for reading and writing spatial representations of visited places in GU.

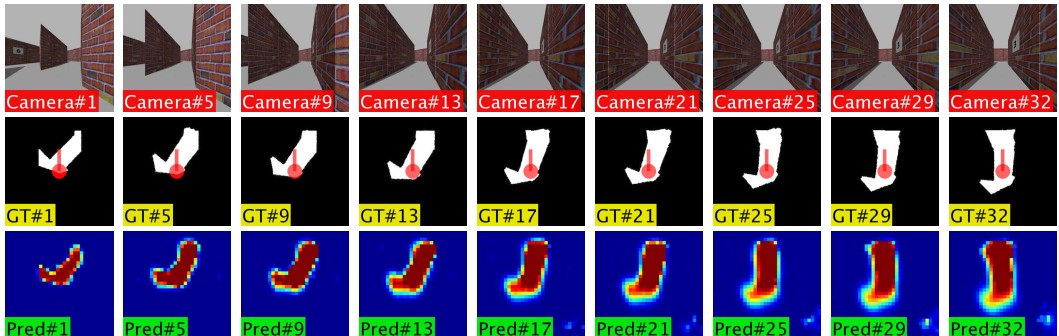

Figure 9: Example results of predicted local maps over first 32 time steps in Maze 6. Frames #1, 5, 9, 13, 17, 21, 25, 29, 32 are shown (left to right columns). The topmost row shows the camera view. Row 2 shows the ground truth with red arrows denoting the agent's position and orientation from the top view. The white region denotes free space while the black denotes unknown areas. Row 3 shows the corresponding top-view accumulative belief of the predicted local maps where the red color denotes higher belief of the free space. Best viewed in color.

| current camera view | estimated egomotion | predicted local map at previous time step |
|---|---|---|
| nn.Sequential() | nn.ParallelTable() | |
| nn.SpatialConvolution(3,128, 4,4, 2,2, 1,1) | nn.AffineGridGeneratorBHWD(A, A) | nn.Reshape(batchSize, 1, 32, 32) |
| nn.ReLU(true) | | nn.Transpose({3,4},{2,4}) |
| nn.SpatialConvolution(128,256, 4,4, 2,2, 1,1) | nn.BilinearSamplerBHWD() | |
| nn.SpatialBatchNormalization(256,1e-3) | nn.Transpose({2,4},{3,4}) | |
| nn.ReLU(true) | nn.Reshape(batchSize, 32, 32) | |
| nn.SpatialConvolution(256,512, 4,4, 2,2, 1,1) | nn.ReLU() | |
| nn.SpatialBatchNormalization(512,1e-3) | | |
| nn.ReLU(true) | | |
| nn.SpatialConvolution(512,1024, 4,4, 2,2, 1,1) | | |
| nn.SpatialBatchNormalization(1024,1e-3) | | |
| nn.ReLU(true) | | |
| nn.SpatialFullConvolution(1024,512, 4,4, 2,2, 1,1) | | |
| nn.SpatialBatchNormalization(512,1e-3) | | |
| nn.ReLU(true) | | |
| nn.SpatialFullConvolution(512,256, 4,4, 2,2, 1,1) | | |
| nn.SpatialBatchNormalization(256,1e-3) | | |
| nn.ReLU(true) | | |
| nn.SpatialFullConvolution(256,128, 4,4, 2,2, 1,1) | | |
| nn.SpatialBatchNormalization(128,1e-3) | | |
| nn.ReLU(true) | | |
| nn.SpatialFullConvolution(128,1, 4,4, 2,2, 1,1) | | |
| nn.SpatialBatchNormalization(1,1e-3) | | |
| nn.ReLU(true) | | |
| nn.Squeeze() | | |
| nn.View(batchSize, 32,32) | | |
| nn.CAddTable() | | |
| nn.HardTanh() | | |

Table 6: Architecture of Boundary Vector Unit (BVU)

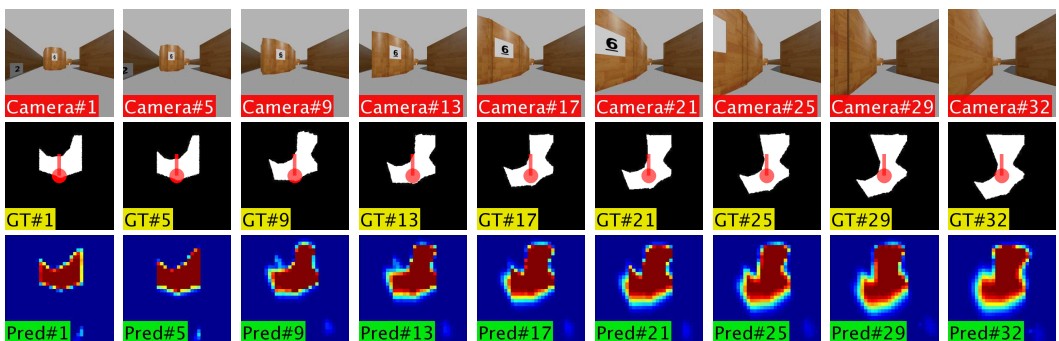

Figure 10: Example results of predicted local maps over first 32 time steps in Maze 7. Frames #1, 5, 9, 13, 17, 21, 25, 29, 32 are shown (left to right columns). The topmost row shows the camera view. Row 2 shows the ground truth with red arrows denoting the agent's position and orientation from the top view. The white region denotes free space while the black denotes unknown areas. Row 3 shows the corresponding top-view accumulative belief of the predicted local maps where the red color denotes higher belief of the free space. Best viewed in color.

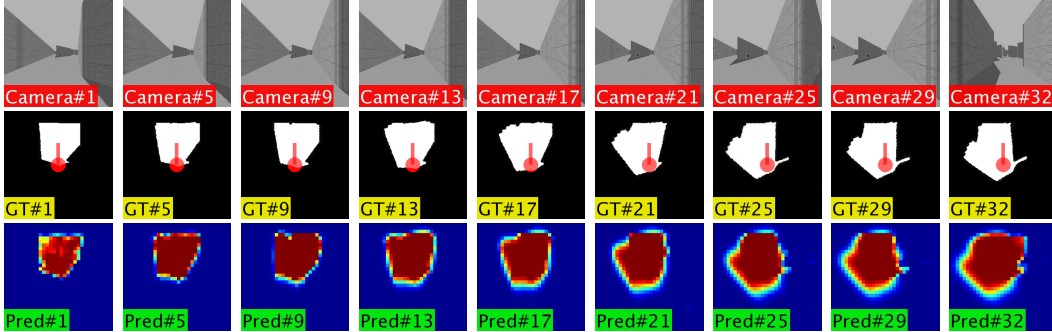

Figure 11: Example results of predicted local maps over first 32 time steps in Maze 8. Frames #1, 5, 9, 13, 17, 21, 25, 29, 32 are shown (left to right columns). The topmost row shows the camera view. Row 2 shows the ground truth with red arrows denoting the agent's position and orientation from the top view. The white region denotes free space while the black denotes unknown areas. Row 3 shows the corresponding top-view accumulative belief of the predicted local maps where the red color denotes higher belief of the free space. Best viewed in color.

| contacted two RGB camera views at $t$ and $t-1$ |
| :---: |
| nn.Sequential() |
| nn.SpatialConvolution(6,128, 4,4, 2,2, 1,1) |
| nn.ReLU(true) |
| nn.SpatialConvolution(128,256,4,4, 2,2, 1,1) |
| nn.SpatialBatchNormalization(256,1e-3) |
| nn.ReLU(true) |
| nn.SpatialConvolution(256,512,4,4, 2,2, 1,1) |
| nn.SpatialBatchNormalization(512,1e-3) |
| nn.ReLU(true) |
| nn.SpatialConvolution(512,1024,4,4, 2,2, 1,1) |
| nn.SpatialBatchNormalization(1024,1e-3) |
| nn.ReLU(true) |
| nn.SpatialConvolution(1024,1024,2,2, 2,2, 0,0) |
| nn.SpatialBatchNormalization(1024,1e-3) |
| nn.ReLU(true) |
| nn.Reshape(1024) |
| nn.Linear(1024,512) |
| nn.ReLU() |
| nn.Linear(512,256) |
| nn.ReLU() |
| nn.Linear(256,6) |
| nn.CrossEntropyCriterion() |

Table 7: Architecture of Head Direction Unit (HDU)

| Anchor | Positive | Negative |
|---|---|---|
| (share the same parameters) | | |
| nn.Sequential() | | |
| nn.SpatialConvolution(3,64, 4,4, 2,2, 1,1) | | |
| nn.SpatialBatchNormalization(64,1e-3) | | |
| nn.ReLU(true) | | |
| nn.SpatialConvolution(64,64, 3,3, 1,1, 1,1) | | |
| nn.SpatialBatchNormalization(64,1e-3) | | |
| nn.ReLU(true) | | |
| nn.SpatialMaxPooling(2,2,2,2,0,0) | | |
| nn.SpatialConvolution(64,128, 4,4, 2,2, 1,1) | | |
| nn.SpatialBatchNormalization(128,1e-3) | | |
| nn.ReLU(true) | | |
| nn.SpatialConvolution(128,128, 3,3, 1,1, 1,1) | | |
| nn.SpatialBatchNormalization(128,1e-3) | | |
| nn.ReLU(true) | | |
| nn.SpatialMaxPooling(2, 2, 2, 2, 0, 0) | | |
| nn.SpatialConvolution(128,256, 3,3, 1,1, 1,1) | | |
| nn.SpatialBatchNormalization(256,1e-3) | | |
| nn.ReLU(true) | | |
| nn.SpatialConvolution(256,256, 3,3, 1,1, 1,1) | | |
| nn.SpatialBatchNormalization(256,1e-3) | | |
| nn.ReLU(true) | | |
| nn.SpatialMaxPooling(2, 2, 2, 2, 0, 0) | | |
| nn.SpatialConvolution(256,512, 3,3, 1,1, 1,1) | | |
| nn.SpatialBatchNormalization(512,1e-3) | | |
| nn.ReLU(true) | | |
| nn.SpatialConvolution(512,512, 3,3, 1,1, 1,1) | | |
| nn.SpatialBatchNormalization(512,1e-3) | | |
| nn.ReLU(true) | | |
| nn.SpatialMaxPooling(2, 2, 2, 2, 0, 0) | | |
| nn.View(batchSize, -1) | | |
| nn.Linear(4*512,512) | | |
| nn.ReLU(true) | | |
| nn.Linear(512,128) | | |
| nn.ReLU(true) | | |
| nn.NarrowTable(1,2) | nn.NarrowTable(2,2) | |
| nn.PairwiseDistance(1) | nn.PairwiseDistance(1) | |
| nn.DistanceRatioCriterion(true) | | |

Table 8: Architecture of Place Unit (PU)

| all the past predicted global maps,
the predicted local map at current time step,
the predicted local map at the time step when the place unit recognizes the previously visited place |
|---|
| nn.Sequential() |
| nn.VolumetricConvolution(n_channel,128, 4,4,4, 2,2, 2, 1,1,1) |
| nn.VolumetricBatchNormalization(128,1e-3) |
| nn.LeakyReLU(0.2, true) |
| nn.VolumetricConvolution(128,256, 4,4,4, 2,2, 2, 1,1,1) |
| nn.VolumetricBatchNormalization(256,1e-3) |
| nn.LeakyReLU(0.2, true) |
| nn.VolumetricConvolution(256,512, 4,4,4, 2,2, 2, 1,1,1) |
| nn.VolumetricBatchNormalization(512,1e-3) |
| nn.LeakyReLU(0.2, true) |
| nn.VolumetricConvolution(512,1024, 4,4,4, 2,2, 2, 1,1,1) |
| nn.VolumetricBatchNormalization(1024,1e-3) |
| nn.LeakyReLU(0.2, true) |
| nn.VolumetricFullConvolution(1024,512, 4,4,4, 2,2,2, 1,1,1) |
| nn.VolumetricBatchNormalization(512,1e-3) |
| nn.LeakyReLU(0.2, true) |
| nn.VolumetricFullConvolution(512,256, 4,4,4, 2,2,2, 1,1,1) |
| nn.VolumetricBatchNormalization(256,1e-3) |
| nn.LeakyReLU(0.2, true) |
| nn.VolumetricFullConvolution(256,128, 4,4,4, 2,2,2, 1,1,1) |
| nn.VolumetricBatchNormalization(128,1e-3) |
| nn.LeakyReLU(0.2, true) |
| nn.VolumetricFullConvolution(128,1, 4,4,4, 2,2,2, 1,1,1) |
| nn.VolumetricBatchNormalization(1,1e-3) |
| nn.LeakyReLU(0.2, true) |
| nn.Reshape(batchSize, seqlength, 64, 64) |
| nn.SpatialConvolution(seqlength,1, 1,1, 1,1, 0,0) |
| nn.MSECriterion() |

Table 9: Architecture of Map Correction

