# OpenReview forum: "Egocentric Spatial Memory Network"
_ICLR.cc/2018/Conference — Reject_

### Official Review · AnonReviewer1 · 2017-11-27
**Complex model, tiny test set--unconvincing results**

**Rating:** 3
**Confidence:** 4

**Review:**

The paper proposes a biologically inspired model of mammalian navigation which includes head direction cells, boundary vector cells, place cells, and grid cells. The proposed model includes modules for all of these kinds of cells and includes: an Neural Touring Machine, Spatial Transformer Network, Recurrent Neural Networks, and CNNs. The model is trained with supervision to output the overhead map of the global map. All components are trained with dense supervision (e.g. loop closure, ego motion with orientation-position, and the ground truth local overhead map). The model is trained on 5 mazes and tested on 2 others.

I believe that this paper is severely flawed. Firstly, the model has ample free parameters to overfit when such a tiny test set is used. Are the test environments sufficiently different from the training ones? For example, when showing that the head direction cells generalize in the new mazes how can we be sure that it is not using a common lighting scheme common to both train and test mazes to orient itself? Also , because MSE is not scale free error measure it is hard to tell how significant the errors are. What is the maximal possible MSE error in these environments?

To quote the authors "However, there is no existing end-to-end neural network for
visual SLAM to our best knowledge." For example "RatSLAM: a hippocampal model for simultaneous localization and mapping" by Milford at al. was a successful biologically inspired SLAM algorithm (able to map neighborhoods using a car mounted monocular camera) first published in 2004--with many orders of magnitude fewer free parameters.

---

### Official Review · AnonReviewer2 · 2017-11-28
**Good paper: an original idea with an effective and reproductible implementation (many details are provided).**

**Rating:** 5
**Confidence:** 4

**Review:**

The paper is well written, well-motivated and the idea is very interesting for the computer vision and robotic communities. The technical contribution is original. The vision-based agent localization approach is novel compared to the methods of the literature. However, the experimental validation of the proposed approach could be more convincing (e.g. by testing on real data, with different training and testing splitting configurations).

Major concern:
1) The authors depict in section 2 “there is no existing end-to-end neural network for visual SLAM to our best knowledge” but they should discuss the positioning with respect to the paper of M Garon and JF Lalonde, “Deep 6-DOF Tracking”, ISMAR 2017 which propose a fully neural network based camera tracking method.

Minor concerns:
2) Table 3: the comparison is not rigorous in the sense that the proposed method estimates a 2D pose (3-DOF) while ORB-SLAM and EKF-SLAM are methods designed for 3D pose estimation (6-DOF). Is it possible to generalize your method to this case (6-DOF) for a more consistent comparison? At least, the fact that your method is more restrictive should be discussed in the paper.

3) In the same vein than point 2), ORB-SLAM and EKF-SLAM are methods based on regression while the proposed method is restricted to the classification pose estimation. Is it possible to test your method with a regression task?

4) It would be interesting to test the proposed method on real data to measure its robustness in terms of noise sensor and in terms of motion blur.

5) It would also be interesting to test the proposed method on datasets usually used in the SLAM community (e.g. using the sequences of the odometry benchmark of KITTI dataset).

6) In the SLAM context, the running time aspect on the test phase is crucial. Hence, the authors should compare the running time of their method with algorithms of literature (e.g. ORB-SLAM).

---

### Official Review · AnonReviewer3 · 2017-11-28
**Significance of Contributions Unclear**

**Rating:** 4
**Confidence:** 4

**Review:**

Significance of Contributions Unclear


The paper describes a neural network architecture for monocular SLAM that is argued to take inspiration from neuroscience. The architecture is comprised of four components: one that estimates egomotion (HDU) much like prediction in a filtering framework; one that fuses the current image into a local 2D metric map (BVU); one that detects loop closures (PCU); and one that integrates local maps (GU). These modules along with their associated representations are learned in an end-to-end fashion. The method is trained and evaluated on simulated grid environments and compared to two visual SLAM algorithms.

The contributions and significance of the paper are unclear. SLAM is arguably a solved problem at the scales considered here, with existing solutions capable of performing localization and mapping in large (city-scale), real-world environments. That aside, one can appreciate the merits of representation learning in the context of SLAM and a handful of neural network-based approaches to SLAM and the related problem of navigation have been proposed of-late. However, the paper doesn't do a sufficient job making the advantages of the proposed approach over these methods clear. Further, the paper emphasizes parallels to neuroscience models for navigation as being a contribution, however these similarities are largely hand wavy and one could argue that they also exist for the many other SLAM algorithms that perform prediction (as in HDU), local/global mapping (as in BVU and GU) and loop closure detection (as in PCU). More fundamentally, the proposed method does not appear to account for motion or measurement noise that are inherent in any SLAM problem and, related, does not attempt to model the uncertainty in the resulting map or pose estimates.

The paper evaluates the individual components of the architecture. The results suggest that the different modules are doing something reasonable, though the evaluation is rather limited (in terms of spatial scale) and a bit arbitrary (e.g., comparing local maps to the ground truth at a seemingly arbitrary 32s). The evaluation of the loop closure is limited to a qualitative measure and is therefore not convincing. The authors should quantitatively evaluate the performance of loop closure in terms of precision and recall (this is particularly important given effects of erroneous loop closure detections and the claims that the proposed method is robust). Meanwhile, it isn't clear that much can be concluded from the ablation studies as there is relatively little difference in MSE between the two ablated models.


Additional comments/questions:


* A stated advantage of this method over that of Gupta et al. is that the agent's motion is not assumed to be known. However, it isn't clear whether and how the model incorporates motion or measurement uncertainty, which is fundamental to any SLAM (or navigation) framework.

* Related, an important aspect of any SLAM algorithm is an explicit estimate of the uncertainty in the agent's pose and the map, however it doesn't seem that the proposed model attempts to express this uncertainty.

* The paper claims to estimate the agent's pose as it navigates, but it is not apparent how the pose is maintained beyond estimating egomotion by comparing the current image to a local map.

* Related, it is not clear how the method balances egomotion estimates and exteroceptive measurements (e.g, as are fused with traditional filtering frameworks). There are vague references to "eliminating discrepancies" when merging measurements, but it isn't clear what this actually means, whether the output is consistent, or how the advantages of egomotion estimation and measurements are balanced.

* The BVU module is stated as generating a "local" map, but it is not clear from the discussion what limits the resulting map to the area in the vicinity of the robot vs. the entire environment.

* It is not clear why previous data is transformed to the agent's reference frame as a result of motion vs. the more traditional approach of transforming the agent's pose to a global reference frame.

* The description of loop closure detection and the associated heuristics is confusing. For example, Section 3.4 states that the agent only considers positions that are distant from the most recent visited position as a means of avoiding trival loop closures, however Section 3.4 states that GU provides memory vectors near the current location for loop closure classification.

* The description of the GU module is confusing. How does spatial indexing deal with changes to the map (e.g., as a result of loop closures/measurement updates) or transformations to the robot's frame-of-reference? What are h, H, w, and W and how are they chosen?

* The architecture assumes a discrete (and course) action space, whereas actions are typically continuous. Have the authors tried regressing to continuous actions or experimenting with finer discretizations that are more suitable to real applications?

* It is not clear what is meant by the statement that the PU "learns to encode the representation of visited places".

* The means by which the architecture is trained is unclear. What is the loss that is optimized? How is the triplet loss (Eqn. 3) incorporated (e.g., is it weighted differently than other terms in the loss)?

* Section 3.2 states that the "agent has to learn to take actions to explore its surroundings", however it isn't apparent that the method reasons over the agent's policy. Indeed, this is an open area of research. Instead, the results section suggests that the agent acts randomly.

* Section 4.1 draws comparisons between HDU and Head Direction Cells, however the latter estimate location/orientation whereas the former (this method) predicts egomotion. While egomotion can be integrated to estimate pose (as is done in Fig 4), these are not the same thing.

* The authors are encouraged to tone down claims regarding parallels to navigation models from neuroscience as they are largely unjustified.

* The comparison to existing monocular SLAM baselines is surprising and the reviewer remains skeptical regarding the stated advantages of the proposed method. How much of this difference is a result of testing in simulation? It would be more convincing to compare performance in real-world environments, for which these baselines have proven effective.

* Figure 1: "Border" --> "Boundary"

* Figure 1: The camera image should also go to the BVU block

* Many of the citations are incorrectly not parenthesized

* The paper should be proof-read for grammatical errors

---

### Decision · Program_Chairs · 2018-01-29
**ICLR 2018 Conference Acceptance Decision**

**Decision:**

Reject

**Comment:**

Authors do not respond to significant criticism - e.g. lack of a critical reference
Reviewers unanimously reject.